# Improving realty management ability based on big data and artificial intelligence decision-making

**Aichun Wu** *

School of Management Engineering, Guangxi Construction Vocational and Technical College, Nanning, China

* wu_aichun@outlook.com

## Abstract

Realty management relies on data from previous successful and failed purchase and utilization outcomes. The cumulative data at different stages are used to improve utilization efficacy. The vital problem is selecting data for analyzing the value incremental sequence and profitable utilization. This article proposes a knowledge-dependent data processing scheme (KDPS) to augment precise data analysis. This scheme operates on two levels. Data selection based on previous stagnant outcomes is performed in the first level. Different data processing is performed in the second level to mend the first level's flaws. Data processing uses knowledge acquired from the sales process, amenities, and market value. Based on the knowledge determined from successful realty sales and incremental features, further processing for new improvements and existing stagnancy mitigation is recommended. The stagnancy and realty values are used as knowledge for training the data processing system. This ensures definite profitable features meeting the amenity requirements under reduced stagnancy time. The proposed scheme improves the processing rate, stagnancy detection, success rate, and training ratio by 8.2%, 10.25%, 10.28%, and 7%, respectively. It reduces the processing time by 8.56% compared to the existing methods.

## 1. Introduction

### 1.1 Realty management

Realty or property management is a process that supervises an organization's or industry's properties. Realty management is an important task that manages properties based on priorities and commitments. The systems manage legal disputes and interests of the properties [1]. Realty Management provides various amenities to the users that provide effective services. The property management system maintains, controls, oversees and processes real estate or physical properties. The market value of the properties is also maintained by management systems [2]. Various techniques and technologies are used for realty management systems. Cloud-based techniques are most widely used for property management systems. Cloud computing identifies the flexible solutions required to maintain the properties [3, 4]. A cloud computing

**Data Availability Statement:** All the datas are within the manuscript.

**Funding:** This paper is supported by 2021 young and middle-aged teachers' basic scientific research ability promotion project "Research and practice on

risk prevention and control of property service enterprises from the perspective of the civil code" (Project No.: 2021KY1144). The funders had no role in study design, data collection and analysis, decision to publish, or preparation of the manuscript.

**Competing interests:** The authors have declared that no competing interests exist.

system detects the economic scale of a property and provides necessary information for management systems. Machine learning (ML) based methods are used for property management systems. The support vector machine (SVM) algorithm is usually used as an ML technique that improves accuracy in real-time management systems. SVM predicts the functions and operations required to provide better performance in the management process [5]. Location, economy, government policy, and consumer preferences may impact real estate markets. Adding domain knowledge and context-specific elements to the learning process is paramount.

Cloud computing is a significant component of real estate management since it facilitates providing computer services via the internet. This technology enables data storage, processing, and analysis on remote servers. This technology allows for the retrieval and examination of large amounts of organized and unorganized data from many sources, such as property listings, market trends, and consumer behaviour. Artificial intelligence (AI) can be used in the real estate industry to automate property valuations, optimize pricing tactics, and uncover investment opportunities. This is achieved using algorithms that can learn from data and generate predictions or suggestions. Machine learning algorithms, which are a subset of artificial intelligence (AI), demonstrate exceptional proficiency in identifying patterns and may be trained using past data to make predictions about future trends in the real estate market, anticipate property prices, and provide recommendations for suitable renovations or improvements to optimize returns. An example of this would be a machine learning model trained using historical sales data. This model would be capable of analyzing various attributes of a property and its surrounding area to estimate its prospective market worth. This would greatly assist real estate agents and investors in making well-informed judgments.

## 1.2 Importance of big data in realty management

Big data extraction is a process that extracts data from other systems or sources. Big data extraction identifies the important data and variables required for further application processes. The big data extraction method provides feasible information to execute a certain system task. Big data extraction is used for realty management systems. Big data is mainly used to decrease the complexity and latency in the data identification progression [6]. Big data analytic techniques identify the optimal information from a huge amount of data. Big data extraction first analyzes the database containing relevant data for property management systems [7]. Big data extraction extracts important features from other systems, improving the systems' flexibility and reliability range. Users widely use Internet of Things (IoT) based property management systems. IoT uses various methods that predict the contents for further processes. Commercial real estate (CRE) contains important operational functions. Big data extracts data from CRE that reduces the complexity of realty management systems [8, 9]. The efficacy and acceptance of the big data and AI decision-making system are directly affected by how user-friendly the interface is for realty managers to engage with it. The ease of use, data entry, result interpretation, and decision-making for realty managers is guaranteed by an intuitive interface. Realty management decision-making is revolutionized by integrating big data and artificial intelligence (AI) technology. These improved capabilities boost efficiency, accuracy, and strategic insights.

The quality, completeness, and diversity of real estate data might vary. The accuracy and reliability of the insights obtained by artificial intelligence models might be compromised by using inaccurate or inadequate data. A decision-making framework incorporating big data and AI in realty management aims to leverage large volumes of diverse data to inform decision-making processes through advanced analytics and machine learning. Improving realty management capabilities with big data integration entails making better decisions, running

more efficient operations, and managing assets more effectively by harnessing massive amounts of varied, complicated, and real-time data. Evaluating and measuring realty management abilities often entails looking at several aspects or criteria to see how well real estate management operations work. The ability to competently and efficiently manage real estate assets, properties, and transactions is known as realty management ability. Maximizing profits and ensuring the optimum use of real estate assets include a variety of operations linked to the purchase, development, operation, and disposal of real estate.

## 1.3 Importance of artificial intelligence in realty management

Artificial intelligence (AI) is widely utilized in different fields and applications. AI technique is primarily utilized to enhance the performance and efficiency of the systems. AI is used in the decision-making process of realty management systems [10]. Decision-making is a complicated task to execute in the realty management system. The decision-making process requires accurate datasets to make the proper decision [11]. AI-based decision-making algorithm is used in realty management. Important key values and characteristics are identified by AI using the feature extraction technique [12]. Feature extraction extracts the variables for decision-making, decreasing the computation process's latency. The AI-based algorithm improves overall decision-making accuracy, enhancing the effectiveness of property management systems [13]. Artificial neural network (ANN) is utilized for the decision-making process. ANN analyzes the data that are required for decision-making. ANN identifies the key features from the database. ANN decreases the latency and complexity in the computation procedure, enhancing the decision-making process's feasibility range [14, 15]. Big data handling and processing with precision for reliable property management through diverse considerations requires complex computations. With the help of big data and AI, property managers can sort through data peaks and make informed decisions. Advanced analytics and machine learning algorithms may uncover patterns, trends, and insights that are not obvious through conventional approaches. Implementing stringent restrictions on access following the concept of least privilege. Verify user identities and allocate access rights following task responsibilities. Remain at the forefront of the company's ever-evolving demands by regularly reviewing and updating access privileges. A decision-making framework incorporating big data and AI in realty management aims to leverage large volumes of diverse data to inform decision-making processes through advanced analytics and machine learning. Real estate market trends may be foretold with AI algorithms by analyzing past data. The accuracy of property evaluations is ensured by AI algorithms considering many parameters, such as property attributes, market circumstances, and previous sales data. Cleansing, validating, and enriching data are all part of strong data quality assurance procedures. To guarantee high-quality datasets for analysis, this study will use AI techniques to manage incomplete or noisy data.

The issue of enhancing real estate management proficiency emerges within real estate management, wherein proficient management of property-related data is of paramount importance. Multiple parties, such as property owners, real estate agents, property managers, and tenants, are impacted by this matter. Efforts have been undertaken to address this issue by advancing diverse data processing methodologies and technology. One method is implementing knowledge-dependent data processing schemes (KDPS) to improve real estate data management by utilizing domain-specific knowledge and norms. In the processing and analysis of property-related data, these schemes employ many techniques, including rule-based systems, expert systems, and ontology-based approaches, to collect and apply domain knowledge effectively. Furthermore, researchers have also investigated advancements in fields such as machine

learning and artificial intelligence to enhance the precision and effectiveness of real estate data processing.

## 1.4 Contribution of this study

Designing and implementing a Knowledge-Dependent Data Processing Scheme (KDPS) for realty management requires careful attention to accessibility, privacy, security, ethical implications, and other concerns. The performance metrics include stagnancy detection, success rate, training ratio, and reduced processing time. Several key performance indicators (KPIs) and metrics may be used to measure and evaluate the progress in realty management capabilities, especially with installing a Knowledge-Dependent Data Processing Scheme (KDPS). The primary objective of the knowledge-dependent data processing scheme (KDPS) is to optimize real estate management by utilising domain-specific knowledge and data analysis methodologies. Data acquisition and organization encompass collecting and arranging pertinent information from diverse sources, including property listings, market trends, and client preferences. The data is further subjected to processing and analysis utilizing knowledge-based rules and algorithms that integrate domain experience derived from specialists in the real estate industry. The insights that have been retrieved are utilized to construct prediction models and decision support systems. These systems are designed to aid in property assessment, investment analysis, and client targeting tasks. By integrating data-driven insights with human experience, KDPS empowers real estate managers to make well-informed decisions, resulting in greater operational efficiency, optimized resource allocation, and improved customer satisfaction. The assessment should consider qualitative and quantitative metrics to provide a full picture of the effect. Real estate managers can foresee changes in the market because of data on market trends and economic factors such as inflation and interest rates. The value of a property may be better understood by looking at its sales history.

Considering its significance, this article introduces a knowledge-based processing scheme that is split and briefed as given below:

1. Introducing a knowledge-based processing scheme for validating property data upon multiple criteria such as market value, maintenance, and sales

2. Incorporating knowledge learning for improving the operations of realty systems towards flawless data handling and precise recommendations

3. Performing an extensive analysis using appropriate external data to verify its adaptability for the proposed scheme

The rest of the paper is organized as follows. Section 2 describes the various researchers opinions regarding the realty systems. Section 3 discusses the working process of the Knowledge-Dependent Data Processing scheme-based realty management process and the system's efficiency evaluated in section 4. Conclusion described in Section 5.

## 2. Literature review

### 2.1 Deep learning-based models for realty management

Wang et al. [16] developed a DL model for house price prediction utilizing heterogeneous information analysis. The proposed model is a joint self-attention model that extracts necessary data from the database. The price prediction process evaluates the availability of amenities, rapid transit, location, and property environment. The proposed model identifies the important datasets that are required for prediction. The suggested model attains high accuracy

in prediction, improving the systems' performance. However, the system requires additional effort to analyse a large data volume.

Chen et al. [17] introduced a semiconductor intellectual property (SIP) business model for value creation analysis. The key objective of the introduced model is to evaluate the actual value creation mechanism for SIP firms. The firms' intellectual property (IP) is identified, producing optimal data for business models. The introduced model classifies the IP based on specific characteristics and priorities. The SIP business model is mostly used for owners' entrepreneurship. The study offers insights into semiconductor intellectual property's economic models and dynamic capacities. However, its generalizability may be limited due to its special focus on the semiconductor industry. The generalizability of the findings and consequences to industries subject to distinct dynamics, technical landscapes, and intellectual property considerations may be limited.

Su et al. [18] proposed building information modelling (BIM) and machine learning (ML) integration framework automated property evaluation. An information extraction method is used here to identify important information from the database. ML mainly detects the key values required for the evaluation process. The suggested framework enhances the information exchange process among individuals. Numerical outcomes demonstrate that the proposed framework decreases the error ratio in the evaluation process. The potential amalgamation of Building Information Modeling (BIM) and machine learning methodologies for automated property valuation presents encouraging outcomes. However, it is important to acknowledge that the study's scope may be constrained by the accessibility and reliability of BIM data and the intricate nature of real-world situations that our models may not comprehensively encompass. Furthermore, the precision and dependability of the valuation models may be impacted by the particular datasets employed for training and validation.

## 2.2 Decision-making models for realty management

Hu and Lin [19] designed a multi-criteria group decision-making methodology (MCGDM) for property management systems under a z-number environment. The suggested technique is a novel aggregation method that decreases the complexity of management systems. MCGDM identifies the problems during decision-making that reduce the latency in the computation process. The suggested MCGDM increases the decision-making process's accuracy, enhancing the systems' performance range. The practical implementation of the proposed integrated multi-criteria group decision-making methodology for property concealment risk assessment may face challenges due to the intricate nature of collecting dependable and precise Z-number data, encompassing both fuzzy and probabilistic uncertainties. Furthermore, the efficacy of the methodology may be impacted by the proficiency and subjective assessments of the decision-makers engaged in the collective decision-making process.

Syed and Lawryshyn [20] introduced a multi-criteria decision-making technique for the risk analysis process. Important key values and Risk Matrix (RM) are identified from the database, producing appropriate data for further decision-making. A cost-benefit analysis is utilized here to analyze the appropriate data required for the systems. RM produces accurate datasets, which reduce the complexity and latency in the risk assessment process. The introduced method reduces the risk issues, increasing the network efficiency and effectiveness. Due to its complexity and processing overhead, the suggested multi-criteria decision-making approach may hinder its practical applicability in real-time or time-sensitive scenarios.

Fu and Hu [21] proposed a binary tree pricing method using a machine learning (ML) technique for farmland management systems. A strategy gradient reinforcement learning algorithm is implemented in the pricing technique to analyze the important features of the

database. The proposed method uses basic principles which produce feasible conditions for the analysis process. Compared with other approaches, the proposed technique increases the financial criteria and performance of the networks. However the method consumes high computation complexity.

Lee et al. [22] introduced an adaptive decision-making method for civil infrastructure systems. The primary objective of the introduced technique is to evaluate the risks that occur during infrastructure systems. The introduced method identifies the dynamic changes in the communication and interaction process. Community risks and issues are identified, providing feasible data for decision-making. The introduced method achieves high accuracy in decision-making but consumes high computation time.

Anejionu et al. [23] developed a cloud-enabled big data infrastructure for spatial urban data systems (SUDS). The developed method uses an analytic technique that analyzes SUDS' important characteristics and features. Variations of urban keys and values are identified from the database, decreasing the computation process's latency. The real-time data source provides relevant data for the analytics process. Simulation outcomes illustrate that the suggested technique increases prediction accuracy, which improves SUDS's performance range. However, the method requires optimization approach while handling large volume of data.

Zhang and Carter [24] proposed a grazing system model for industries. The proposed model is an online information system that provides various services and functions to the users. Remote sensing techniques are used here to capture sensible data using wireless sensors. Wireless sensors reduce the difficulty of the computation process. The objective of the recommended model is to identify climate change in grazing lands. The recommended model enhances the overall effectiveness ratio of grazing land systems. Although, the training system requires additional effort to manage the scalability.

Zhang et al. [25] developed a data-driven decision strategy for offshore assets. A multicore classification model is utilized here that recognizes the exact information types based on the conditions. The data mining method is also used here to detect important datasets for decision-making. The multicore classification model improves offshore assets' feasibility and computation efficiency. The developed model maximizes the accuracy in decision-making, enhancing the systems' efficiency and reliability range.

März [26] introduced a new decision-making method for small private landlords (SPL). Both theoretical and practical aspects of the data, which produce relevant data for various processes, are identified. SPL gathers information from the low-demand housing market, providing feasible data for decision-making. Certain privacy policies and schemes are used in SPL, enhancing the systems' performance range. The presented method attains high accuracy in decision-making, high the computation process's complexity range.

Teresa [27] developed a racialized class-monopoly rent for land contracts. The primary objective of the proposed technique is to offer proper security services to the owners. The suggested technique identifies the vulnerabilities and issues that occur during rent. The identified effects provide relevant data for security policies. The proposed method creates various services via the class-monopoly process. The suggested approach maximizes the security and privacy range in the renting process. Although requires the standard security techniques to ensure the privacy on large volume of data.

Peng [28] proposed a machine-learning assessment method for hazard shock. The key objective of the suggested technique is to determine property and land prices for the decision-making process. Disaster notification and alert messages that provide important characteristics and features for various processes are analyzed. Key features of hazard effects and shock are also identified, reducing the computation process's latency. The suggested technique increases the performance and efficiency level of the systems.

A study by Zhou et al. [29] proposed using AI to aid decision-making in the great data age. The study begins by surveying the relevant articles from the International Journal of Information Management (IJIM) to provide a historical perspective of AI. Following this, the article delves into AI's decision-making capabilities in general and the challenges of integrating and interacting with AI to supplement or replace human decision-makers. This article offers twelve research proposals to IS researchers concerning conceptual and theoretical development, AI technology-human interaction, and AI implementation. These proposals aim to promote the study of AI decision-making in the Big Data age.

Foster Provost and Tom Fawcett presented data science, big data, and data-driven decision-making [30]. Conceptual frameworks, which are an integral aspect of data science, facilitate this. Automated pattern extraction from data, for instance, follows a standard procedure. By breaking down problem-solving into its component parts, one may make it more methodical, less error-prone, and easier to understand. Big data technology, data science methods grounded on big data, and data-driven decision-making have a strong track record of significantly boosting company performance.

Kaile Zhou et al. [29] suggested using big data for smart energy management. A four-pronged approach is covered: demand-side management (DSM), asset management and collaborative operation, power generation-side management, and microgrid and renewable energy management. After that, we'll look at the evolution of smart energy management inside industries powered by big data and talk about it. In conclusion, the author stresses the difficulties of smart energy management powered by big data in information technology infrastructure, data gathering and governance, data integration and sharing, processing and analysis, privacy and security, and qualified individuals.

The data analytics-based technique for strategic decision-making was introduced by Murat Ozemre and Ozgur Kabadurmus [31]. This research uses a large quantity of publicly available trade data to predict future export quantities using two distinct machine learning algorithms: Random Forest (RF) and Artificial Neural Networks (ANN). The BCG Matrix, used for strategic market analysis, incorporates the predicted values. A made-up case study of a Chinese firm that exports freezers and refrigerators is used to verify the suggested technique. Based on the findings, it is clear that the suggested approach facilitates efficient strategic market research and produces reliable trade predictions. Regarding the accuracy of forecasts, the RF outperforms the ANN as well.

Business strategy alignment and big data analytics capability (BDAC) were both addressed by Akter [32] as means to enhance company performance. The BDAC model is suggested in this work by combining the entanglement perspective of sociomaterialism with the resource-based theory (RBT). Using an entanglement conceptualization of the higher-order BDAC model and its effect on FPER, the results validate its usefulness. The findings also show how analytics capability-business strategy alignment significantly moderates the BDAC—FPER connection.

The relationship between big data analytics and the performance of firms was discussed by Samuel Fosso Wamba et al. [33]. This study adds to the existing literature by investigating the impact of business process architecture on firm performance (FPER) and how process-oriented dynamic capabilities (PODC) mediate this connection. The results confirm the importance of the hierarchical BDAC model's entanglement conceptualization, which directly and indirectly affects FPER. The findings back up PODC's substantial mediation function in boosting FPER and insights. The author discusses what this means for future studies and clinical work.

Big data analytics' strategic business value was proposed by VARUN GROVER [34]. By building on previous work on the value of IT, this research provides a framework for

understanding BDA, and it then uses real-world examples of BDA to show how this framework works. Then, we review the framework's capabilities in studying BDA value generation and realization-focused components and linkages. The author takes a problem-oriented approach to the framework, arguing that issues with BDA components could spark specific concerns for further investigation. Research and practice in BDA may be better targeted via the strategic use of data resources if this study is structured in a way that allows for the development of a substantial research agenda.

Organizational decision-making might benefit from the Human-AI Symbiosis, as proposed by Mohammad Hossein Jarrahi [35]. Humans can still provide a more comprehensive, intuitive approach to handling ambiguity and uncertainty in organizational decision-making, but AI may augment human cognition when it comes to complicated problems via increased computational information processing capability and an analytical approach. The concept of "intelligence augmentation" is similar to this premise: artificial intelligence technologies should complement human work, not replace it.

Healthcare Big Data and the Medical Internet of Things were proposed by Dimiter V. Dimitrov [36]. The MIoT is essential to healthcare's digital transformation because it paves the way for new business models, facilitating process modifications, productivity gains, cost reduction, and improved customer experiences. These days, exercise, health education, symptom monitoring, care coordination and collaborative illness management are all made possible with smartphone applications and wearable technology. Users will spend less time making sense of data outputs thanks to platform analytics that make data interpretations more relevant. Discoveries made possible by analyzing massive amounts of data will propel the digital revolution in healthcare, corporate operations, and real-time decision-making.

Implementing big data analytics and business intelligence was pioneered by Hsinchun Chen et al. [37]. As a result of the severity and breadth of data-related issues facing modern businesses, business intelligence and analytics (BI&A) has grown in prominence among academics and industry professionals alike. Using almost a decade of relevant academic and business publications, the author presents the results of a bibliometric analysis of influential BI&A articles, scholars, and study subjects. At last, the six papers that make up this special issue are presented and described according to the BI&A research concept.

Based on the survey, there are several problems with existing methods in attaining a high processing rate, including stagnancy detection, success rate, training ratio, and reduced processing time. Hence, this study introduces a knowledge-based processing scheme (KPDS) for validating real estate property data based on market value, maintenance, and sales criteria.

The dataset from [38] validates the proposed scheme's performance. This data is acquired from various real estate systems and their logs. The rental, sales, maintenance, and mortgage data are validated through 18 classes, 4931 entries, and 323894 events. Therefore, the first analyses are presented to justify the need for KDPS. Fig 1 presents the average maintenance expenses between 2014 and 2022 for rental and sales.

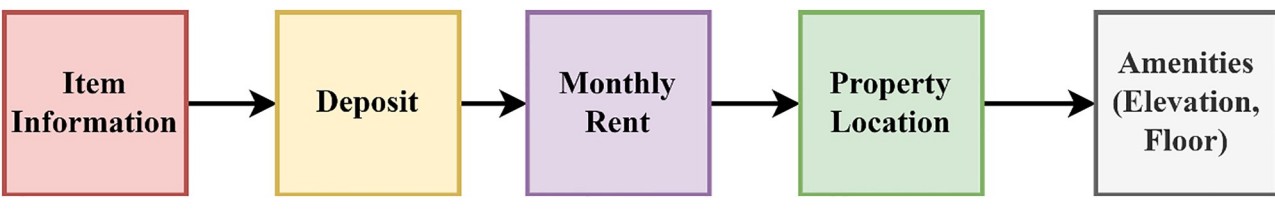

**Fig 1. Property analysis (Source: Self).**

Fig 1 representations reflects the property components of reality management item description, monthly rental, amenities, unit area with building floor count, size and location. Indicators of the financial condition of these properties include the amount of the deposit and the monthly rent, while specifics such as the number of rooms, the unit space, and the number of floors in the building provide information on the buildings' physical characteristics. It is also possible for the existence of amenities such as elevators and storage facilities to have an impact on the popularity and value of the aforementioned properties.

Using context-aware information, users may better comprehend the recommendations and their relevance to the larger realty management objectives. Developing user-friendly interfaces that let people investigate and query the model's decision-making process. Users may better understand the underlying variables and examine particular recommendations with the help of interactive tools. Incorporating domain-specific information and experience into the data analysis process is a limitation addressed by the knowledge-dependent data processing scheme (KDPS). The utilization of human expertise and decision-making rules allows the system to manage limitations in data and surmount computational intricacies effectively. The versatility of KDPS enables its application in several industries, effectively overcoming the challenge of limited generalizability through integrating industry-specific knowledge and data sources.

It is recommended that the KDPS be improved by integrating real-time data streams from diverse sources, including property listings, market trends, and economic indicators. The constant integration of real-time data into the system makes the analytical and decision-making processes grounded in the most current information. Furthermore, it is recommended that the KDPS incorporate methods to facilitate prompt updates and model retraining, enabling the system to adjust swiftly to evolving market conditions. One potential approach is to implement periodic or event-driven updates, wherein the knowledge base and decision rules are revised in accordance with the most recent data and market conditions. By overcoming this constraint, the KDPS may offer more precise and pertinent observations, empowering real estate experts to make well-informed choices that accurately represent the present condition of the market.

## 3. Knowledge-dependent data processing scheme

Realty management process management is done based on the data determined from the prior successful and failed results from purchase to utilization. This process also depends on sales, maintenance, and reports. The cumulative data at these stages are used to improve the utilization efficiency. The Knowledge Dependent Data Processing Scheme (KDPS) is introduced to develop accurate data analysis. Big data is defined as large data generated in high volume, variety, and velocity, which needs new processing techniques to enable better decision-making, knowledge detection, and process expansion. Knowledge-based learning is acquiring the required data before the next learning phase. These abilities can be developed by doing the best way to improve via practice and error. Flaw detection is identifying errors and reconstructing the original, error-free data. Data processing is the collection and direction of digital data to produce meaningful information for the realty management process. Data processing is used as information processing, modifying present information in any way the user can identify. Data processing depends on the knowledge acquired to reduce stagnancy. Indeed, KDPS may face challenges due to the ever-changing and unpredictable real estate markets, which might experience sudden shifts during economic instability or unforeseen events. The KDPS model may not be updated in real-time as it depends on historical information to construct its knowledge base. The inability to promptly replace the

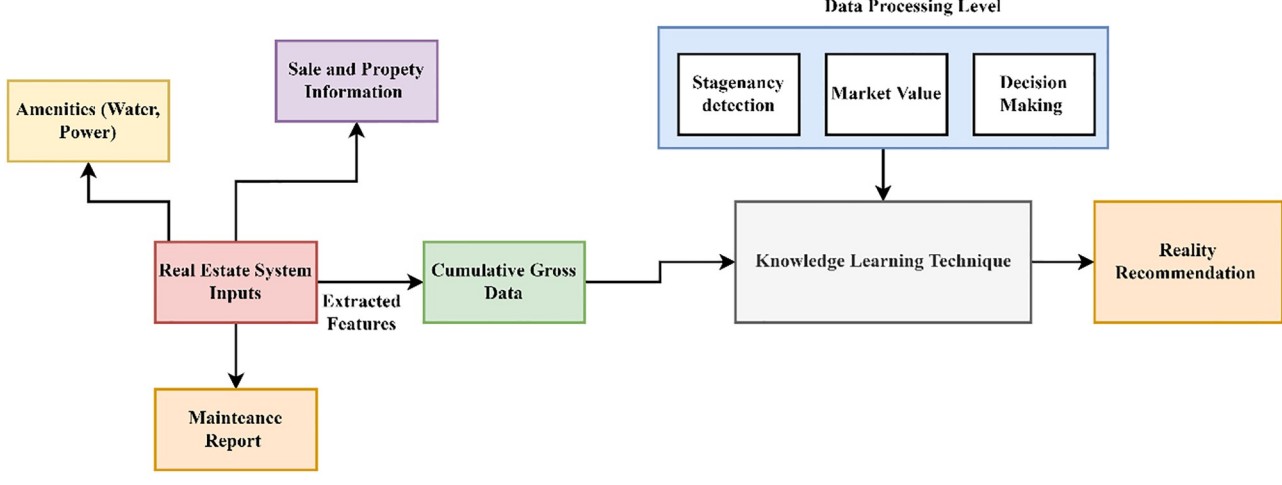

**Fig 2. KDPS model (Source: Self).**

dataset with up-to-date market information might cause the scheme to miss recent developments, resulting in inadequate recommendations. Fig 2 presents the model of the proposed scheme.

In this scheme, the realty management process is done, and the features are extracted to get the gross data. The sales, amenities, and the data reports information are determined. This information determines the cumulative gross data for the processing levels. Based on the cumulative big data output, the processing levels are done with the help of the knowledge-learning technique. This scheme operates on two levels. Data selection based on previous stagnant results is performed in the first level. In the second level, recognizable data processing is accomplished to rejuvenate the first level's flaws. The data processing is sent into two levels to check for improvements and stagnancy. These outputs will be given to the knowledge identification based on the market value. If the value is increased, the prior market value is maintained correctly. However, if it is decreased, market values are not maintained properly, and some recommendations are to be made to improve them. These recommendations are made to overcome the stagnancy and enhance maintenance to reach the proper market value. The flaw and real-time outcomes are used as knowledge for training the upcoming data processing system. This provides the definite operation of the manufacturing process, which meets the real-time requirements under reduced flaws. The knowledge learning technique does these processing levels, and thus, it also helps reduce the flaws in the processing. This method uses the realty management process to improve its features. Because real estate markets are ever-changing, the relative value of certain attributes may shift over time. For a KDPS to adjust to changing market circumstances, it must consider temporal dynamics or update its feature set appropriately. The chosen features must be comprehensive and of high quality. The KDPS can learn patterns from inaccurate or inadequate data during feature engineering, which might reduce its performance in real-world situations. The sales, amenities, and data report information are extracted based on the improvements. The sale information has all the data on the real property and its state, whether it can sell or not, and the profit and loss of the sale. The economic level of the property can be determined in this sale information. The sale can also be increased by improving the maintenance of the process and the facilities of the real property. The procedure of obtaining the sale information of the property can be elucidated by the

subsequent Eq (1) given below:

$$\hat{a}(b) = 1 + \xi + \Sigma_i a_i b(t - iT_s) \tag{1}$$

Where $\hat{a}$ is signified as the information of the realty management, $b$ is indicated as the sale information of the process, $\xi$ is represented as the state of the process data, $T_s$ is symbolized as the features of realty management. Now, the amenities information for the property management is determined. In this, the maintenance of the process is identified, and the facilities of the process are determined. Here, the amenities are identified as whether they have the required facilities to maintain the process or if there is any need to improve the facilities. If there is any need to enhance the process, the information will be helpful in the improvement process by giving the proper data. There should be frequent maintenance of the realty management, and also, there must be the required facilities for better sales. It can enhance the process and reduce flaws. The subsequent Expression (2) describes the process of determining the amenities information from the real operation (2) provided below:

$$S(t) = S[(1 + \xi)V(u, t) + \Sigma_i V(u, t)a_i b(t - iT_s)] + n(u, t) \tag{2}$$

Where $S(t)$ is denoted as the amenities information, $V$ is represented as the facilities of the realty management, $u$ is elucidated as maintenance of the process, $n$ is represented as the process of amenity checking. Now, the data reports are extracted from the realty management process pursued. This data report has the combined data of the sales and amenities information. This will have information about the sales of the process and also the maintenance and facilities of the process. From this data, improvements can be made in the processing levels, and thus, from this output, the big data can be formed for further processes. The data reports have the entire process information; thus, its output will be sent to the cumulative big data for further processing. The process's value and characteristics can be obtained from this information, and its procedure can also be elucidated in this data report. This can be used to improve the processing rate in a shorter time. The data collected from the sale and the amenity collaborates into single information as the data reports. This will have a good sensing operation for the production of successful outcomes. The subsequent Eq (3) clarifies the process of extracting the data reports from the realty management (3) provided below:

$$\left.\begin{array}{l} f(I) = \dfrac{1}{\sqrt{2\pi}\sigma I} \sum_i^n \left( \dfrac{-(I - \eta)^2}{2\Pi^2} \right) \\[3mm] f(J) = \dfrac{1}{\sqrt{2\pi}\sigma J} \sum_j^n \left( \dfrac{-(J + /\sigma^2/2)^2}{2\Pi^2} \right) \end{array}\right\} \tag{3}$$

Where $f(I)$ is denoted as the sales data, $f(J)$ is denoted as the amenities data, $\pi$ is represented as the process of estimating two data, $\sigma$ is signified as the data reports. These data's outputs are joined to form the cumulative gross information. The cumulative data at different stages are used to improve production and delivery efficiency. The exploration of improving realty management ability using big data and artificial intelligence (AI) decision-making can be prompted by several factors, including data volume and capacity, predictive analytics, automation, market dynamics and risk mitigation. Big data is the large data originating in high information gained from the data reports, amenities, and sales, which needs new processing techniques to enable better decision-making, knowledge detection, and process expansion. Concerns about accountability and trust might arise from a lack of transparency, particularly in a field where decisions can have substantial financial consequences. Performing impact

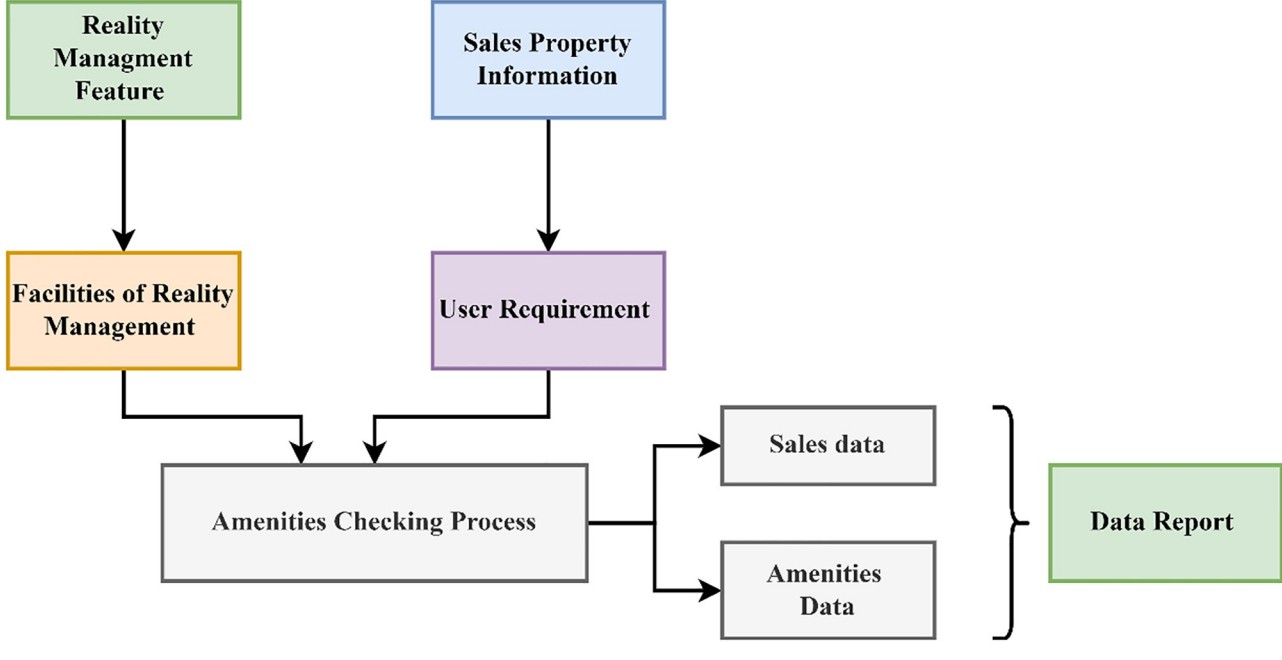

**Fig 3. Gross data validation for amenities checking (Source: Self).**

assessments to determine how KDPS recommendations will play out in practice. This requires checking that the model complies with all applicable laws and ethical guidelines and evaluating how it will affect various stakeholders. Gross cumulative data reduces processing levels and its flaws and stagnancy. This data contains large and complex datasets to be imparted with the data processing application. Big data is linked with a large body of data we cannot comprehend when utilized only in lesser amounts. The output of the cumulative gross data is sent to the processing level, which is done using the knowledge-learning technique. The gross data-based amenities checking is presented in Fig 3.

The cumulative gross data combines the output of the sales, amenities, and data reports information. The output of the big data is sent to the processing levels to detect stagnancy and flaws in the process. This will be used for data processing and to reduce the stagnancy or flaws in the process. Identifying and correcting dataset outliers via effective outlier detection algorithms. Clustering, statistical approaches, or machine learning models developed for outlier identification may be used. Preprocessing the data using noise reduction methods helps lessen the effect of background noise and irrelevant data. The FCM algorithm clusters the data points based on fuzzy membership values, allowing each point to belong to multiple clusters with varying degrees of membership. It can effectively handle missing data by incorporating fuzzy membership values for missing features, enabling the imputation of missing values based on the cluster assignments. One way to make the KDPS more resilient is to use smoothing algorithms or remove unnecessary data. With this outcome, the efficiency of the data processing process can be improved, and the recommendation rate can be reduced due to stagnancy. This processing can be done using the knowledge learning technique on two levels: improvements and stagnancy reduction (Fig 3). The optimal combination of batch size and real-time processing for optimizing computational effectiveness will depend on the requirements. The accruable data at distinct approaches enhance origination and carting capability. The process of getting

the cumulative gross data from the sales, amenity, and data reports is described by the subsequent Eq (4) provided below:

$$
\left.
\begin{aligned}
f(V/S_0) &= \frac{1}{\xi} fI\left(\frac{V}{\xi}\right) * fN(V) \\
&= \int_0^\alpha \frac{1}{\sqrt{2\pi}\sigma x} \Sigma_1^n \left(\frac{-\left(\frac{x}{\xi} + \frac{\sigma^2}{2}\right)}{2\pi^2}\right)^2 \\
&= \frac{1}{\sqrt{2\pi}\sigma P} \Sigma_i^n \left(\frac{-(V - x^2)}{2\sigma^2 P}\right) dx
\end{aligned}
\right\}
\tag{4}
$$

Where $S_0$ is denoted as the cumulative gross data, $N$ is expressed as the output of the sales, amenity and the data reports information. Now, this output will be sent to the data processing procedure. Data processing is performed using the knowledge obtained from the production process, plan, and outcome. The output of the gross data is sent for processing and checking its improvements and stagnancy level. Realty data processing involves a large amount of input data given by the gross data with the amount of output. It is a form of information processing that will alter the information detected by big data. In this data processing procedure, the validation of the realty management, aggression, analysis, and reporting of the management process will be checked. From the output, the improvement of the realty management will be checked. The market value will be referred to for the checking process. The data selection will be done using knowledge-learning techniques based on the previous stagnant results for this process. This input may also help enhance the upcoming realty management process. A knowledge-dependent data processing system (KDPS) success relies on taking realty market changes into account throughout time. The scheme's adaptability to shifting patterns over time may be enhanced by including time-series analysis and techniques for dynamic model updates.

Data processing with the gross data output will also be helpful in the modification operations after the verification procedure. The data will be processed at the processing level based on the values. Data processing in realty management is used to validate the sales and amenities of the process. The data acquired is used to check the stagnancy in the process, and thus, further steps are taken to reduce the flaws. This output helps the upcoming processes without stagnancy and with less processing time. This processing procedure can detect and reduce all the cumbersome and inefficiency. It also maintains property management sustained by learning technology. Data processing can be done to check and identify flaws in the process. The knowledge determined from the sales, amenities, and market value will support processing information. The data processing is based on the complete output of the sales processes and cumulative big data information for profitable utilization. Here, the data selection is also done to analyze the value increments and stagnancy reduction. The data processing operation is used to improve the previous outcome and eliminate the flaws. This can be based on the data reports information extracted from the realty management. It plays a vital role in furthering the upcoming realty management process and improving its value without error. The subsequent Equation elucidates the procedure of data processing

(5) provided below:

$$f(V/S_1) = \frac{1}{2 + \xi} fI\left(\frac{V}{\xi}\right) * fN(V)$$

$$= \int_0^\alpha \frac{1}{\sqrt{2\pi\sigma x}} \Sigma_1^n \left(\frac{-\left(\frac{x}{2+\xi} + \frac{\sigma^2}{2}\right)}{2\pi^2}\right)^2$$

$$= \frac{1}{\sqrt{2\pi\sigma P}} \Sigma_i^n \left(\frac{-(V - x^2)}{2\sigma^2 P}\right) dx$$

(5)

Where $S$ is denoted as the data provided, now it sends to check its improvements based on the value in the processing levels. The knowledge learning technique is used in this processing procedure to check its improvements and stagnancy based on the values obtained by the process. Based on the values obtained by the previous processes, the present values determined will be compared. After the comparisons, the improvement can be identified. The data processing procedure results in some improvements and stagnancy, and then it will be checked whether it is improved compared to the previous process. The upgrades and the stagnancy will be verified based on the knowledge determined from the successful realty management sales and incremental features. To measure how much each feature contributed to the KDPS predictions, feature importance analysis should be performed. Permutation importance, information gain, or Shapley Additive exPlanations (SHAP) values may help us understand the most essential features. The stagnancy and realty management values are used to learn about the data processing system. If there is any lag in the improvements, further effective training will be given to the data processing. The learning for improvements recommendation through verification is illustrated in Fig 4.

The improvements in the data processing will be checked based on the values, and the data will be given needed training if there are no improvements. The output of the gross data is used to verify the enhancement and the stagnancy of the realty management. This

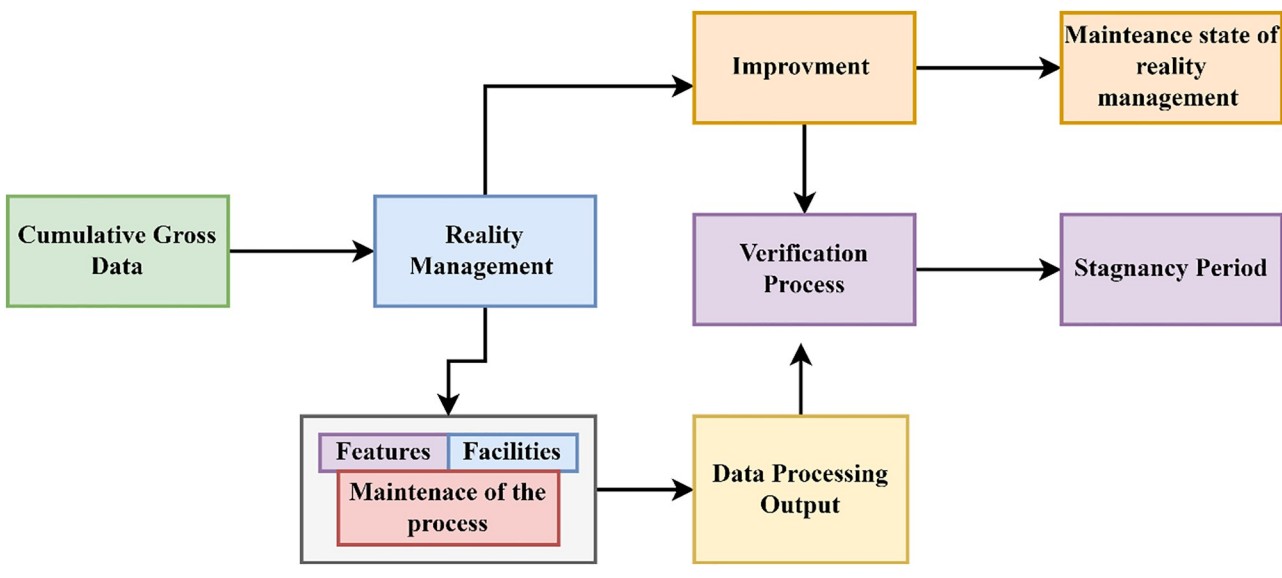

**Fig 4. Improvements recommendation through verification (Source: Self).**

improvement checking procedure depends on the knowledge of the sales, amenities, and information reports. Then, the output is united to form the cumulative gross information, which will help process the information. This is the first level in the procedure based on the previous stagnant outcomes and some steps taken to reduce the stagnancy in the processes (Fig 4). Through thorough training programs, educate users about the KDPS's features, advantages, and limits. Positioning the benefits of AI-based decision-making to real estate management front and centre while addressing any misunderstandings or concerns about the technology. Evaluation of the scheme's ability to adapt external data sources to its internal structure via transformation and preprocessing. Integrating varied datasets into the KDPS is made simple by its capacity to manage data transformations. The process of verifying the improvements in the processing of data is described by the succeeding Eqs (6) and (7) provided below:

$$\left. \begin{array}{l} L_F = \int_{T_{Sh}}^{\alpha} f(V/S_0)dr = \int_{T_{Sh}}^{\alpha} \frac{1}{\xi} fI\left(\frac{V}{\xi}\right) * fN(V)dr \\ \\ L_M = \int_{\alpha}^{T_{Sh}} f(V/S_1)dr = \int_{T_{Sh}}^{\alpha} \frac{1}{2+\xi} fJ\left(\frac{V}{2+\xi}\right) * fN(V)dr \end{array} \right\} \tag{6}$$

$$\left. \begin{array}{l} K_e = (1 - K_1)KF + K_1 K_M \\ \\ = (1 - K_1)\sum_1^n \left(\frac{-\sigma^2}{2}\right) \int_0^{\alpha} \frac{\sqrt{\xi}}{x} \sum_1^n \left(\frac{\frac{-x}{\xi}}{2\sigma^2}\right) \\ \\ = Q(\sqrt{\gamma}(T_{Sh} - x)dx + \int_0^{\alpha} \frac{\sqrt{2+\xi}}{x} \\ \\ = \sum_1^n \left(\frac{\frac{-x}{2+\xi}}{2\sigma^2}\right) B(\sqrt{\gamma}(x - T_{Sh}))dx \end{array} \right\} \tag{7}$$

Where $L_F$ is signified as the output of the data processing, $L_M$ is indicated as the improvements in realty management, $K_e$ is represented as the verification process, $Q$, $\gamma$ is indicated as the previous values. Now, the stagnancy will be checked in the output of the realty management processing based on the knowledge. Stagnancy occurs when the realty management is not maintained well on its amenities. The knowledge of the data report helps identify the stagnancy in the procedure and, during this stagnancy period, helps improve the process with required training. Based on this stagnant outcome, the data is selected for further operation. In this stagnancy period, the performance of the data is verified, and such training is given to reduce the flaws during the procedure. Based on the sales and amenity information, the stagnancy can be identified, and steps can be taken to eliminate the flaws in the upcoming realty management processes. Dealing with missing data is essential to developing robust knowledge-dependent data processing systems (KDPS). If incorrectly handled, missing values in real-world datasets might affect how well models work. Focusing on how various imputation techniques affect KDPS performance using sensitivity studies. Choosing stable and reliable procedures after considering how different approaches to missing data management impact model predictions. The procedure of identifying the stagnancy based on the outcome is

represented by the Eqs (8) and (9) given below:

$$\sum\nolimits_{y\to\alpha} fN(V) = \tau(V) \tag{8}$$

$$\left.\begin{aligned}
\sum\nolimits_{y\to\alpha} Z_F &= \int_{T_{Sh}}^{\alpha} \frac{1}{\xi} f\, I\left(\frac{r}{\xi}\right) dr \\
&= 1 - F_1\left(\frac{T_{Sh}}{\xi}\right)
\end{aligned}\right\} \tag{9}$$

Where $\tau$ is denoted as the flaws in the process, $\alpha$ is represented as the period of stagnancy, $Z$ is indicated as the maintenance state of the realty management, $r$ is indicated as the procedure of eliminating flaws. The process will now be checked with the market values based on the knowledge acquired from the prior outcomes. If the process value is increased or the same as the market values, it is well maintained without any flaws or stagnancy. If the values are decreased than the market value, then it is considered that it is not well maintained and has some stagnancy in the procedure. Then, based on the knowledge from the successful sales, upcoming processing for new realty management process improvements and present stagnancy mitigation is recommended. For a Knowledge-Dependent Data Processing Scheme (KDPS) to work optimally and be durable in many configurations, it is crucial to study how it responds to varied hyperparameter values. Hyperparameter tuning studies aim to find the best values for the hyperparameters by methodically testing them. The recommendations are given to the upcoming process for successful sales values without any flaws; the process of verifying the procedure value with the market value is clarified by the succeeding Eqs (10) and (11) provided below:

$$\left.\begin{aligned}
\sum\nolimits_{y\to\alpha} Z_e &= \sum\nolimits_{y\to\alpha} (1-z_1)Z_F + z_1 Z_M \\
&= (1-z_1) W\left(\frac{T_{Sh} - \xi + \sigma^2/2}{\sigma}\right) \\
&= z_1 W\left(\frac{(2+\xi) - T_{Sh} - \sigma^2/2}{\sigma}\right)
\end{aligned}\right\} \tag{10}$$

$$D(\alpha,\beta,\gamma,\lambda) = \left(\frac{\alpha\beta\gamma}{\lambda}\right)^{\alpha} \frac{\varphi(\beta - \alpha)}{\varphi(\alpha+1)\varphi(\beta)} \tag{11}$$

Where $N$ is signified as the output of present process values, $\lambda$ is represented as the increments in the value, $\varphi$ is denoted as the decrement in the value, $\beta$ is indicated as the output of the verification procedure. Based on this verification, recommendations are given for further processing. This recommendation is given to overcome the stagnancy and improve the process's value. This is used to enhance the data processing levels to achieve market value and eliminate the flaws in the procedure. The recommendation is based on the outcome of the realty management sales to improve the procedure and eliminate stagnancy. This helps to increase the processing ratio, success ratio, and stagnancy reduction. The process of providing the recommendations to improve the value is clarified by the subsequent Eqs (12) and (13)

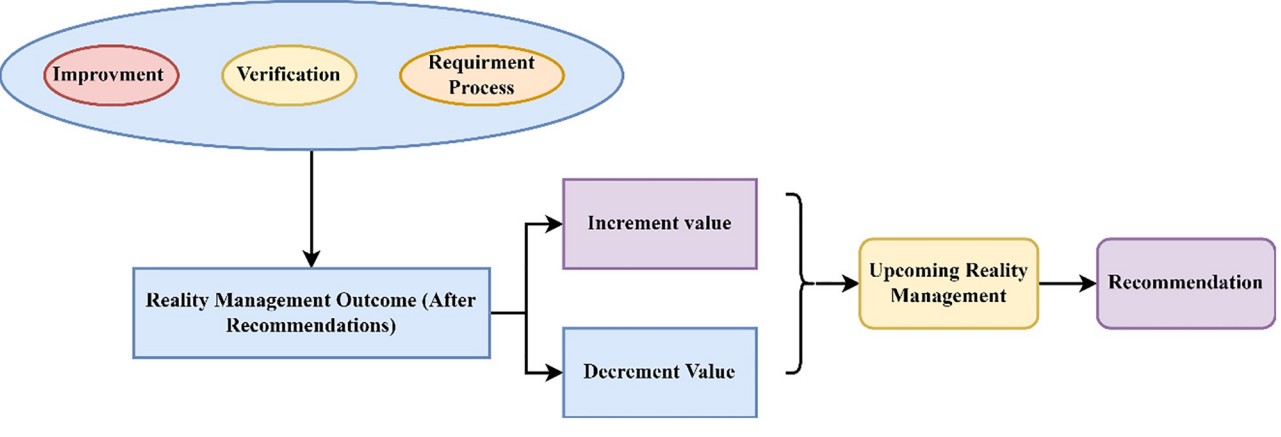

**Fig 5. Recommendation process functions (Source: Self).**

provided below:

$$C_j\left(V, \frac{J}{C}\right) = \sum_{c=0}^{j} \frac{(-1)^c \varphi(V+j+1)}{C!(j-k)!\varphi(V+j-C+1)}\left(\frac{I}{K}\right)^Z \qquad (12)$$

$$O_\varepsilon = \sum_{k=0}^{j}(-1)\frac{j!\varphi(V+1)}{C!(j-k)!\varphi(V+j-C+1)}\mu[j-C] \qquad (13)$$

Where $C_j$ is denoted as the recommendation given to the process, $O_\varepsilon$ is the outcome of the realty management after applying the recommendations. This method uses the knowledge learning technique to improve the data processing levels and eliminate stagnancy. The recommendation process functions are presented in Fig 5. Dividing the dataset into k folds is necessary for k-fold cross-validation for the generalization performance of KDPS. Use the remaining fold to assess the KDPS's performance after training it on k-1 folds. Switching the test fold around every iteration and repeating the procedure k times.

The recommendations are also preferred to enhance the data processing procedure to achieve market value. Here, this method has an increment in the stagnant detection, processing ratio, and success rate. This knowledge learning technique operates on two levels; in the first level, the data selection is done for analysis based on previous stagnant outcomes. In the second level, manifest data operation to eliminate the flaws of the first level procedure. The data processing is performed with the knowledge determined from the production process, sales, amenities, and the gross data outcome. The creation and implementation of KDPS must adhere to well-defined ethical standards and governance methods. Fairness measures should be defined and monitored as part of this process, and the system should be inspected regularly to identify and correct any bias. Making visual representations of the KDPS's decision-making procedures. Some examples of such visual aids that could be useful in explaining how the model works include decision borders, decision trees, or similar structures.

## 4. Discussion

The amenities expected and provided for 412 properties cumulatively for 9 years are required. The incorporation of a complete validation and evaluation methodology, such as k-fold cross-validation, is recommended for the KDPS. The process entails dividing the accessible data into

k subsets, with one subset reserved for testing purposes, while the remaining k-1 subsets are utilized for training the model. The abovementioned procedure is iterated k times, wherein each subset is used as the test set. A more dependable estimation of its capacity for generalization and resilience can be derived by assessing the model's performance across various folds. Furthermore, it is imperative to conduct comprehensive testing of the KDPS on a wide range of real-world and extreme scenarios to evaluate its efficacy across different circumstances. This process entails simulating various market scenarios, property categories, and user specifications to ascertain the system's capacity to accommodate diverse inputs and deliver dependable suggestions or forecasts. In order to verify the validity and robustness of the data analysis, the KDPS utilizes a variety of statistical models and methodologies. Examples of regression models that can analyze the associations between different real estate variables and property values or market trends include linear regression, logistic regression, and ensemble approaches based on decision trees. In order to discover the most relevant models, many model selection strategies are utilized, including cross-validation and information criteria. From the given data, the requirement is extracted with failed maintenance, as presented in Fig 6.

The amenities change with the consumer expectations and the actual requirements over the varying years. From the variations observed between 3, 6, and 9 years span, $\lambda$, and $\psi$ are identified. This serves as the first. $k_e$ for the varying $f(I) + f(J)$ from the extracted data. Based on the analysis, the $V$ improvements from $\sigma$ and $C_j$ are conjoined through balanced average expenses. Therefore, the above requirements rely on $r$ mitigation for $\psi$ and $\lambda$ balancing. If the balancing is achieved, then $n$ checking is performed through various $B$ and $O_\varepsilon$ from the learning process presented in Fig 7.

The $r$ suppression is required over the varying $V$ such that $\psi$ and $\lambda$ are balanced accordingly. The $\lambda$ increments are suppressed through $K_e$ and $L_F$ for $\alpha$ mitigation using $(Q, \gamma)$ such that it relies on $f(I)$, $f(J)$, and $f(I) + f(J)$. Therefore the $L_F$ proceeds with $C_J$ and $T_S$ that is required to reduce stagnancy. Hence $V$, improvements are required for $n(\%)$ reduction. Post this analysis for $\pi$ for $\sigma$ extraction and $C_j$ over the various features presented in Fig 8. To maintain track of modifications to the KDPS codebase, environments, and related files, a version control system like Global Information Tracker (Git) is used. Preserve records of releases and rollbacks using version control.

The $\pi$ is high for $\lambda$ compared to $\psi$ under varying $V$. Compared to the $C_j$, the $n > L_F$ is required for $M$ such that $Z$ is required. This validates the $f(I) + f(J)$ for enhancing the $C_j$ (Fig 8). The KDPS can utilize methodologies such as decision tree visualization, which involves

(a) (b)

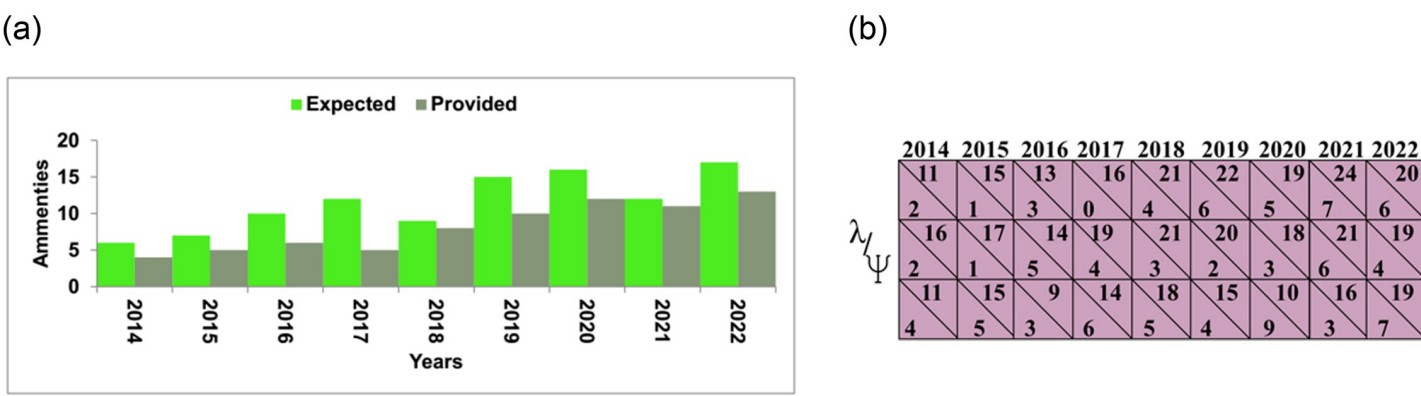

**Fig 6. Amenities (expected and maintenance).**

(a)

(b)

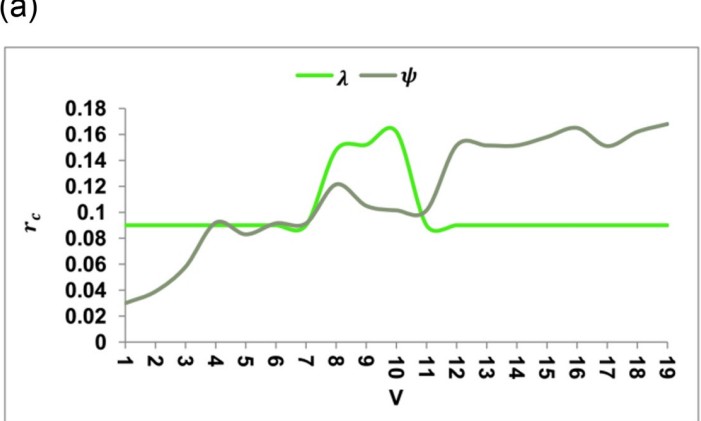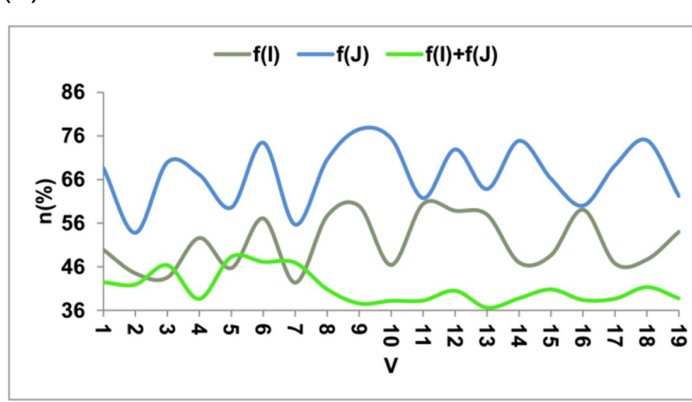

**Fig 7. *r* and *n*(%) analyses.**

presenting the hierarchical arrangement of decision rules and their associated consequences in a user-friendly tree-like structure. Furthermore, implementing decision boundary visualizations allows for the graphical representation of decision borders or regions inside the feature space corresponding to various outcomes or recommendations generated by the KDPS. The following section presents the comparative analysis using data processing rate, stagnancy detection, processing time, success rate, and training ratio. The variants are data inputs (20–260) and amenities (1–15). The additional methods considered are SMMP [25], BIM+AVM [18], and BTPM [21], discussed in the related works section.

### 4.1 Data processing rate

The data processing rate is high in this method, which uses the knowledge-learning technique to improve the values. The data processing is done based on the knowledge determined from the amenities, sales, and data reports. This process is done on two levels; at first, the data will be selected for analysis, and the second level will be used to improve the values of the process. The enhancement and the stagnancy will be checked during this data processing, and further steps will be taken to reduce the flaws. The output of the cumulative gross data also helps in

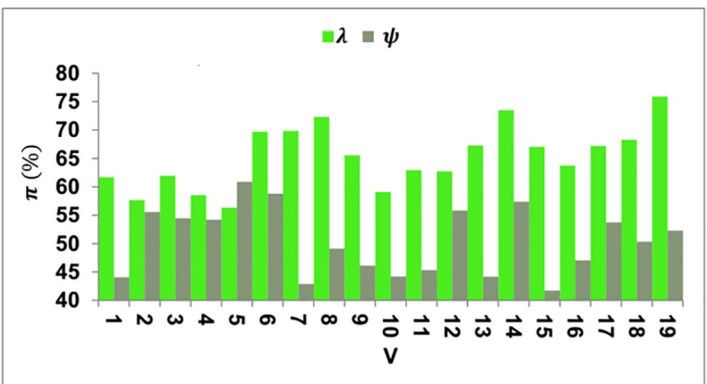

**Fig 8. *π* analysis for *ψ* and *λ*.**

(a)

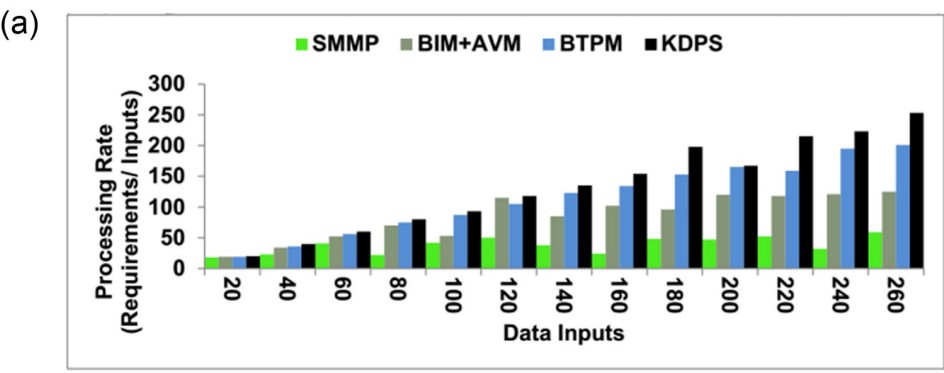

(b)

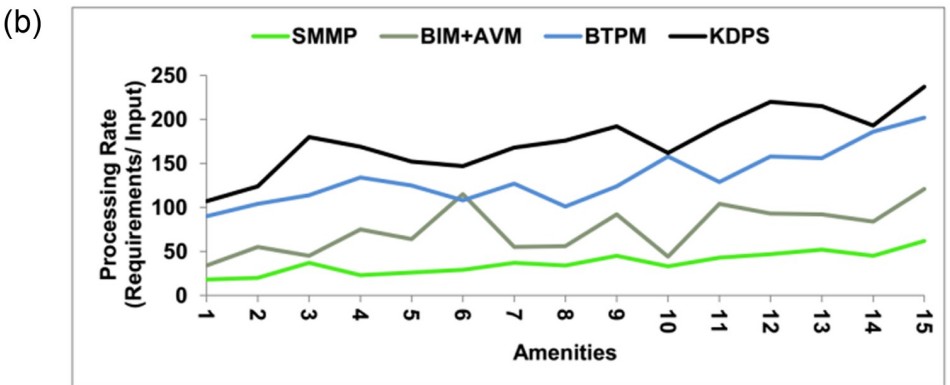

**Fig 9. Data processing rate.**

the processing levels of the data and eliminates the flaws during stagnancy. The output of the process will be verified with the market value to determine its state and whether it is improved or not. If the values are increased, then they are enhanced with good maintenance. If the values are reduced, it has flaws due to the poor amenities, and some steps should be taken to reduce the flaws. Recommendations are given to increase the upcoming data processing values rather than the market values without stagnancy (Fig 9). Feedback should be promptly obtained following interactions to get insights while the user speculates about the experience. The insights derived by real-time or rapid feedback systems are more likely to be correct and relevant.

## 4.2 Stagnancy detection

The stagnancy detection is great in this technique by the learning technique in the processing levels. Based on the outcome of the process, the stagnancy can be determined, and then suggestions can be made to enhance the upcoming realty process. Property stagnation may be influenced by external variables that might change over time, such as neighbourhood development, municipal legislation, and economic situations. The challenge with relying on historical data alone to predict stagnation is that it may not fully capture or forecast these changes. If there is low maintenance amenity and poor sale information, stagnancy occurs in the realty procedure. Stagnancy occurs when the realty process is not maintained better regarding its amenities. The knowledge of data reports helps determine the stagnancy in the procedure, and during this stagnancy period, the necessary training is provided to enhance the process. Based on the stagnancy outcome, the upcoming new reality process will be improved with more efficacious

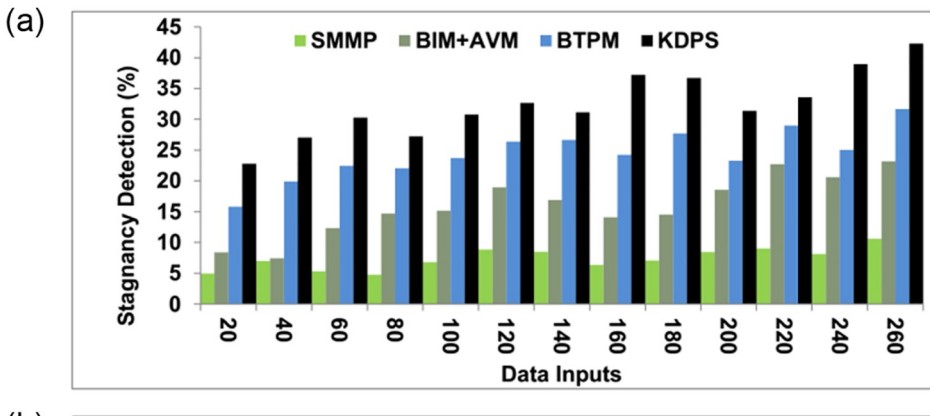

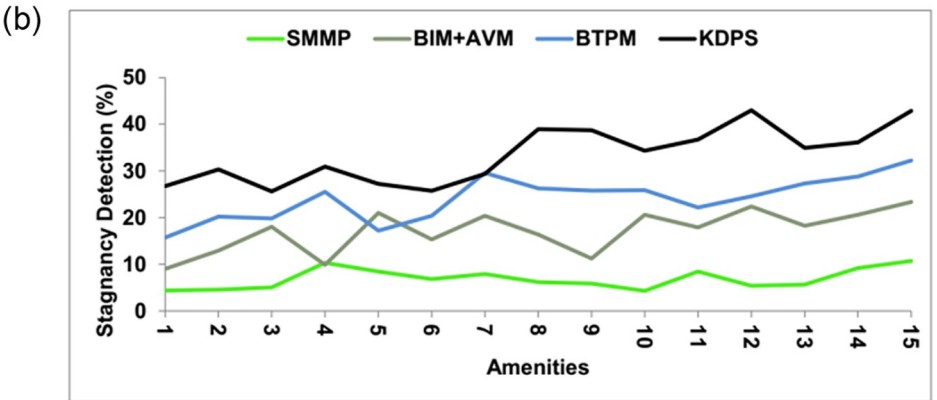

**Fig 10. Stagnancy detection.**

training given to the data. In the stagnancy period, the data are verified for the processing levels, and if there is any lag, then the needed training will be provided to the process to eliminate flaws. Based on the sales and amenity knowledge, the flaws can be detected, and further steps are taken to eliminate the flaws in the upcoming realty management processes (Fig 10).

## 4.3 Processing time

The time taken for processing is less in this method, which utilizes the data processing scheme to enhance the values. The outcome of the cumulative big data helps in the processing levels of the data and checks the stagnancy in the process. From the realty process, the information on the sales, amenities, and data reports are extracted to form the gross information for the successful process. Data processing uses knowledge from production, sales, and previous process outcomes. Further new processing for improvements and flaw mitigation is recommended based on the knowledge determined from successful production and gross data outcome. There should be frequent maintenance of the real process and the required facilities for a better sale. This ensures definite operation of the manufacturing process, meets the real-time requirements and eliminates stagnancies. Effectively processing large-scale data over a cluster of devices is made possible by adopting distributed computing frameworks. As data volumes increase, this KPDS method permits horizontal scalability by introducing more processing nodes to the environment. The sales can also be increased by improving the maintenance of the industrial process and the facilities of the realty process (Fig 11).

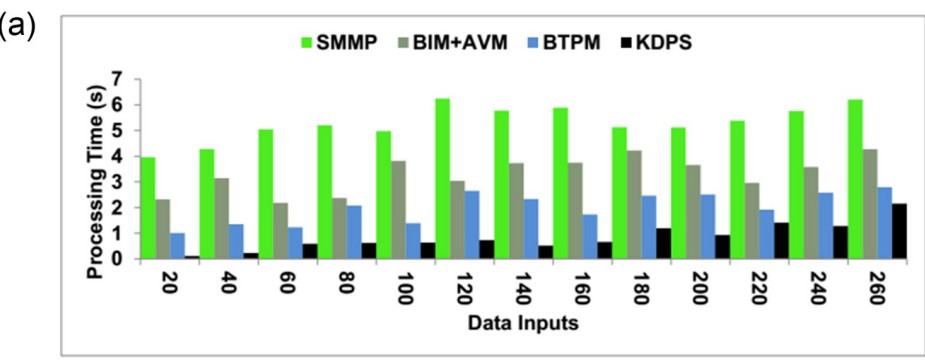

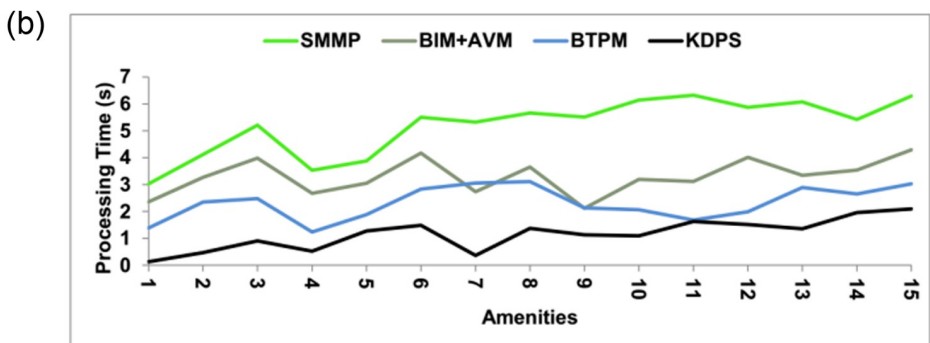

**Fig 11. Processing time.**

## 4.4 Success rate

This method's success rate is better using the processing technique to improve the realty process values. The cumulative data at different stages are used to improve production and delivery efficiency. Big data has large amounts of information originating from high amounts of information gained from data reports, amenities, and sales, which needs new processing techniques to enable better process expansion. Gross cumulative data is used for processing levels to reduce flaws and stagnancy and enhance present values. The market value will be referred to for the verification process. The data selection will be done using knowledge-learning techniques based on the previous stagnant outcomes of this process. This input may also help enhance the upcoming realty management process. This output enables the upcoming process results without the shorter processing time flaws. In this processing procedure, all the cumbersome and inefficiency can be detected and eliminated. Performing bias audits to examine past biases' effect on model predictions systematically. This study analyzes the model's performance across different demographic groupings and pinpoint potential sources of bias for the decision-making process. In this way, the success rate is high in this method (Fig 12).

## 4.5 Training ratio

This method's training ratio is high, and it uses knowledge learning to enhance real-time data processing. After detecting the stagnancy in the process, the needed training is provided to the upcoming process to result in a success rate when compared with the market value. If the output of the verification process is high, then the market value will be such that there will be no need for further training. If the outcome is less than the market value, then training will be

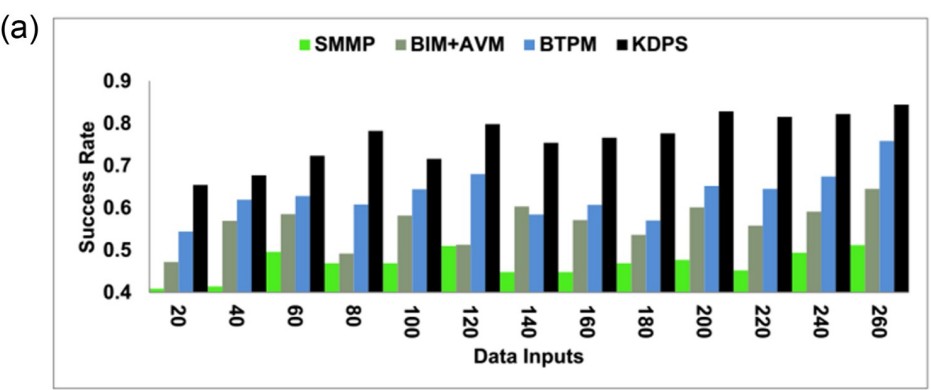

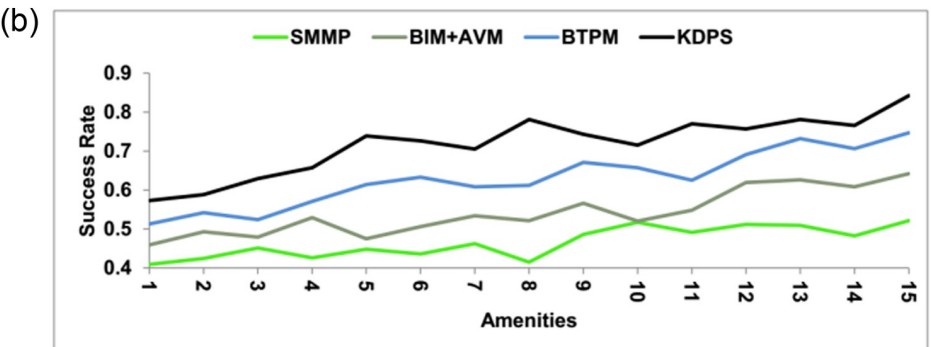

**Fig 12. Success rate.**

given to the upcoming realty processes. A recommendation is given after the market value has been verified. A recommendation is provided to overcome the stagnancy and to improve the realty process value. This is used to enhance the data processing levels to achieve market value and eliminate the flaws in the procedure. Manifest data operation is done to eliminate the flaws of the first realty procedure. This helps to increase the processing ratio, success ratio, and stagnancy reduction. The real estate market is unpredictable and dynamic, with underlying trends that may shift over time. The model could have trouble modifying if the environment changes substantially while the KDPS still assumes stagnant circumstances. The training ratio is high in these techniques with a knowledge learning scheme (Fig 13). Tables 1 and 2 highlight the proposed schemes' performance from the above comparison. Strong security and privacy protocols must be implemented to deal with the delicate nature of real estate data. Building trust with consumers and complying with privacy requirements requires protecting data's confidentiality, integrity, and availability.

## 4.6 Implication of this study

According to the results, the knowledge-dependent data processing scheme (KDPS) demonstrates superior performance compared to the other techniques, specifically SMMP, BIM +AVM, and BTPM, in many real estate management measures. KDPS has the maximum processing rate of 253, which is much greater than the other approaches. On the other hand, SMMP has the lowest processing rate of 59. The KDPS strategy has the largest percentage of cases where the process becomes stagnant or stuck, measuring 42.312%. This indicates that KDPS is more effective than the other techniques in identifying and handling sluggish

(a)

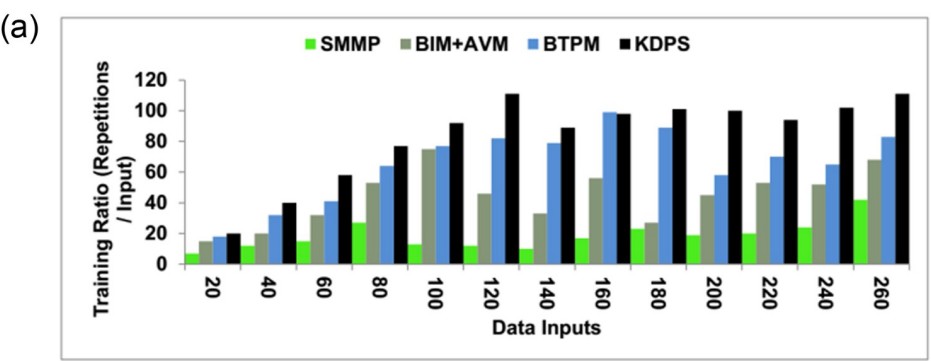

(b)

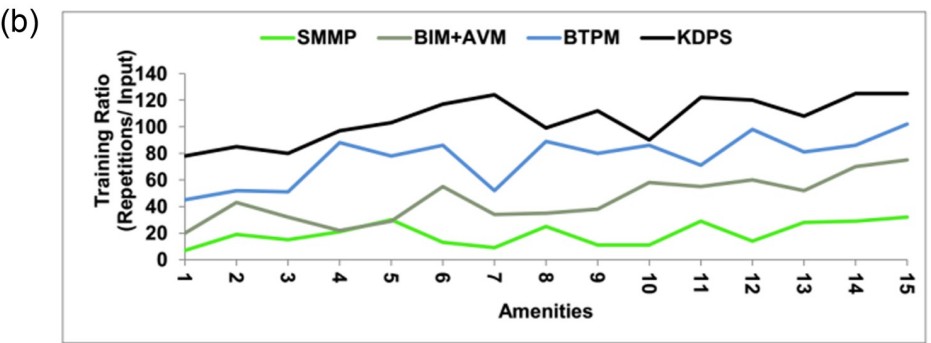

**Fig 13. Training ratio.**

**Table 1. Highlights of performance assessment (data inputs).**

| Metrics | SMMP | BIM+AVM | BTPM | KDPS |
|---|---|---|---|---|
| Processing Rate (Requirements/ Input) | 59 | 125 | 201 | 253 |
| Stagnancy Detection (%) | 10.6 | 23.19 | 31.67 | 42.312 |
| Processing Time (s) | 6.21 | 4.27 | 2.79 | 2.153 |
| Success Rate | 0.512 | 0.645 | 0.758 | 0.844 |
| Training Ratio (Repetitions/ Input) | 42 | 68 | 83 | 111 |

**Findings**: The proposed scheme improves the processing rate, stagnancy detection, success rate, and training ratio by 8.2%, 10.25%, 10.28%, and 7%, correspondingly. It decreases the processing time by 8.56% compared to the existing methods.

**Table 2. Highlights of performance assessment (amenities).**

| Metrics | SMMP | BIM+AVM | BTPM | KDPS |
|---|---|---|---|---|
| Processing Rate (Requirements/ Input) | 62 | 121 | 202 | 237 |
| Stagnancy Detection (%) | 10.72 | 23.34 | 32.22 | 42.848 |
| Processing Time (s) | 6.29 | 4.29 | 3.03 | 2.095 |
| Success Rate | 0.521 | 0.642 | 0.747 | 0.842 |
| Training Ratio (Repetitions/ Input) | 32 | 75 | 102 | 125 |

**Findings**: The proposed scheme improves the processing rate, stagnancy detection, success rate, and training ratio by 7.64%, 10.38%, 10.27%, and 7.38%, correspondingly. It decreases the processing time by 8.96% compared to the existing methods.

situations. KDPS exhibits the shortest processing time, measuring 2.153 seconds, indicating its exceptional efficiency and quickness in managing realty activities. KDPS approach exhibits the highest success rate, as evidenced by its proportion of effective outcomes at 0.844 or 84.4%. This indicates that KDPS outperforms the other methods in generating successful results. The training ratio, which quantifies the number of repetitions or iterations needed per input, is observed to be the highest for KDPS at 111. This suggests that achieving ideal performance may necessitate further training or exposure to data. The findings indicate that the knowledge-dependent data processing scheme (KDPS) demonstrates efficacy and efficiency in the context of realty management tasks. It surpasses alternative methods in multiple metrics, such as processing rate, stagnancy detection, processing time, success rate, and training ratio.

## 5. Conclusion

Realty management is useful in handling diverse types of properties at any distance. The data acquired from different realty events are accumulated periodically for extensive processing. The data precision for analysis is improved through knowledge learning under varying market values, maintenance, and profitable utilization. Based on the knowledge learning, two distinct processes are performed: improvement recommendations and stagnancy analysis. The first processing is performed to meet the demand for amenities and facilities per the consumer's needs. The drawbacks and lacking features associated with realty management are identified in the second process. The identified processes are mitigated through modified recommendations without impacting profitable management. Knowledge of sales, stagnancy, and market values is acquired from verifications and flaw elimination instances. This includes the increments and decrements of the recommendation and conventional realty process. A modular and adaptable design was used to construct the KDPS so that new technology could be easily included without a redesign. Updating and expanding a well-architected system makes it compatible with new technologies. The proposed scheme improves the processing rate, stagnancy detection, success rate, and training ratio by 8.2%, 10.25%, 10.28%, and 7%, respectively. It decreases the processing time by 8.56% compared to the existing methods.

The research conducted by KDPS has the potential to facilitate data-driven decision-making by providing comprehensive market insights, examining patterns to identify risks and opportunities, and facilitating scenario simulations. Resource allocation optimisation is achieved by identifying areas that necessitate attention, strategically targeting marketing activities according to client preferences, and aligning workforce levels with workload projections. Operational efficiency is enhanced by implementing process automation, fostering cross-departmental collaboration through integrating data perspectives and enabling predictive maintenance through data analysis from multiple sources. In addition, KDPS improves client experiences by providing individualized recommendations, implementing focused marketing strategies, and addressing feedback. Furthermore, it supports compliance efforts and proactive risk mitigation methods.

Future research endeavours may prioritize advancing the KDPS by investigating sophisticated knowledge representation methodologies, such as deep learning-based approaches, to capture intricate patterns and correlations in the dataset effectively. Furthermore, the integration of real-time data feeds and the establishment of mechanisms for ongoing learning and adaptation would enhance the ability of the KDPS to deliver decision support that is both fast and accurate in the face of dynamic real estate market conditions.

## Author Contributions

**Funding acquisition:** Aichun Wu.

**Investigation:** Aichun Wu.

**Methodology:** Aichun Wu.

**Resources:** Aichun Wu.

**Writing – original draft:** Aichun Wu.

**Writing – review & editing:** Aichun Wu.

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
