## [Decision Letter · Decision Letter 0]

17 Mar 2024

PONE-D-24-06746Improving Realty Management Ability Based on Big Data and Artificial Intelligence Decision-MakingPLOS ONE

Dear Dr. Wu,

Thank you for submitting your manuscript to PLOS ONE. After careful consideration, we feel that it has merit but does not fully meet PLOS ONE’s publication criteria as it currently stands. Therefore, we invite you to submit a revised version of the manuscript that addresses the points raised during the review process.

We look forward to receiving your revised manuscript.

Kind regards,

Jitendra Yadav, Ph.D.

Academic Editor

PLOS ONE

Journal Requirements:

"2021 young and middle-aged teachers' basic scientific research ability promotion project “Research and practice on risk prevention and control of property service enterprises from the perspective of the civil code” (Project No.: 2021KY1144)"

"2021 young and middle-aged teachers' basic scientific research ability promotion project “Research and practice on risk prevention and control of property service enterprises from the perspective of the civil code” (Project No.: 2021KY1144)"

"2021 young and middle-aged teachers' basic scientific research ability promotion project “Research and practice on risk prevention and control of property service enterprises from the perspective of the civil code” (Project No.: 2021KY1144)"

4. In the online submission form, you indicated that [The datasets used and/or analysed during the current study available from the corresponding author on reasonable request.]. 

6. We note that Figures 1-5 in your submission contain copyrighted images. All PLOS content is published under the Creative Commons Attribution License (CC BY 4.0), which means that the manuscript, images, and Supporting Information files will be freely available online, and any third party is permitted to access, download, copy, distribute, and use these materials in any way, even commercially, with proper attribution. For more information, see our copyright guidelines: http://journals.plos.org/plosone/s/licenses-and-copyright.

a. You may seek permission from the original copyright holder of Figures 1-5 to publish the content specifically under the CC BY 4.0 license. 

Reviewers' comments:

Reviewer's Responses to Questions

**Comments to the Author**

1. Is the manuscript technically sound, and do the data support the conclusions?

Reviewer #1: Partly

Reviewer #2: Partly

Reviewer #3: Partly

2. Has the statistical analysis been performed appropriately and rigorously? 

Reviewer #1: N/A

Reviewer #2: Yes

Reviewer #3: No

3. Have the authors made all data underlying the findings in their manuscript fully available?

Reviewer #1: Yes

Reviewer #2: Yes

Reviewer #3: No

4. Is the manuscript presented in an intelligible fashion and written in standard English?

Reviewer #1: Yes

Reviewer #2: Yes

Reviewer #3: No

5. Review Comments to the Author

Reviewer #1: Thanks for giving me a chance to review this manuscript. This is an interesting topic. The author(s) tries to work significantly. This article proposes a knowledge-dependent data processing scheme (KDPS) to augment precise data analysis. However, still, some of the anomalies I found during the review process are addressed, which may help further develop the study.

(comments are included in the attached file)

Reviewer #2: The manuscript presents a promising contribution to the field of real estate property management, particularly in terms of data-driven analysis and decision-making. However, to achieve its full potential, the study requires substantial revisions to enhance clarity, methodological rigor, and overall presentation. Addressing the outlined recommendations will significantly improve the quality and impact of the research.

The introduction (section 1) lacks clear focus and structure, resulting in a disjointed presentation of ideas. The article jumps between various concepts and technologies without providing a coherent narrative or logical progression of thought. To improve readability and comprehension, the introduction should be organized into distinct sections with clear headings and subheadings to delineate different topics.

The article briefly touches upon various concepts such as cloud computing, big data extraction, AI-based decision-making, and machine learning algorithms without delving into their theoretical underpinnings or practical applications in real estate management. To enhance the scholarly depth of the article, each concept should be discussed in greater detail, providing definitions, explanations, and relevant examples to elucidate their significance in the context of real estate management.

While the article makes several claims about the benefits and efficacy of incorporating AI, big data analytics, and machine learning in real estate management, it lacks empirical evidence or case studies to support these assertions. To strengthen the argument and credibility of the article, empirical validation through case studies, experiments, or real-world examples should be provided to demonstrate the practical utility and effectiveness of the proposed knowledge-dependent data processing scheme (KDPS).

The recommendations provided at the end of the introduction are vague and lack specificity. They mention the introduction of a knowledge-based processing scheme and the incorporation of knowledge learning without elaborating on concrete steps or strategies for implementation. To provide actionable recommendations, the article should offer clear guidelines, methodologies, or frameworks for designing and implementing the proposed KDPS.

In section 1 (Introduction), it’s better to clarify the whole structure of the paper, which can be easier for reader to understand what scenarios have been investigated or examined and what’s main contribution this paper makes. For example, this paper is organized as follows: section 2 present what and section 3 present what… It’s suggested to give a brief overview of entire paper in the introduction.

The article presents a comprehensive list of related works without synthesizing the key findings, trends, or gaps in the existing literature. Instead of merely listing each work, the article should critically analyze the contributions, methodologies, and limitations of previous studies to identify overarching themes or research gaps. Synthesizing the literature would provide readers with a clearer understanding of the current state of knowledge in the field and highlight areas for further investigation.

The article lacks critical evaluation of the methodologies and findings of the referenced works. Each cited study is described briefly without assessing the strengths, weaknesses, or implications of the research. A critical evaluation would involve discussing the methodological rigor, applicability of findings, and potential biases in the studies cited.

While the article provides a comprehensive list of related works, it does not adequately discuss the relevance of each study to the proposed knowledge-based processing scheme (KDPS). The article should clearly articulate how each referenced work informs the development or validation of the KDPS and how it contributes to addressing the identified research problem. Discussing the relevance of previous studies would strengthen the rationale for the proposed approach and demonstrate its novelty or innovation.

One significant limitation is the scheme's reliance on historical data, which may not adequately capture real-time market dynamics. To address this, the KDPS should incorporate real-time data feeds and implement mechanisms for timely updates to adapt to changing market circumstances.

The effectiveness of the KDPS heavily relies on the quality and relevance of the features selected for analysis. There is a need to conduct thorough feature engineering and validation to ensure that the chosen features accurately reflect market trends and dynamics.

Transparency and interpretability are crucial for building trust and understanding in AI-driven decision-making systems. The KDPS should provide clear visual representations of its decision-making processes to stakeholders, such as decision trees or decision borders, to enhance comprehension and trust.

Dealing with missing data is critical for developing robust data processing systems. The KDPS should implement robust strategies for handling missing values, such as imputation techniques, and conduct sensitivity analyses to evaluate the impact of different approaches on model performance.

Optimal hyperparameter values significantly impact the performance of machine learning models. The KDPS should conduct thorough hyperparameter tuning studies to identify the best configuration for its algorithms and ensure optimal performance across different scenarios.

Rigorous validation and evaluation processes are essential for assessing the KDPS's performance and reliability. Implementing techniques such as k-fold cross-validation can provide insights into the model's generalization performance and robustness.

Revise the manuscript to improve clarity by defining technical terms, providing detailed explanations of methodologies, and ensuring logical coherence between sections.

Provide a clear, step-by-step description of data processing and analysis techniques, including any software tools or algorithms utilized. Incorporate references to established methodologies to enhance transparency and reproducibility.

Conduct thorough validation of the proposed approach using benchmark datasets or simulated scenarios. Compare the performance of the proposed method with existing models through quantitative metrics and qualitative evaluation criteria.

Simplify complex concepts and minimize technical jargon to make the manuscript accessible to a broader audience. Use clear, concise language and structured formatting to improve readability and comprehension.

While the conclusion briefly mentions the benefits of the proposed scheme, such as improved processing rate and success rate, it does not delve into the practical implications for real estate management practitioners or stakeholders.

Discussing how the findings can inform decision-making processes, optimize resource allocation, or enhance overall operational efficiency in realty management would provide valuable insights for readers interested in applying the research in practical contexts.

While the conclusion provides a concise summary of the research findings, it could benefit from greater specificity, depth, and discussion of practical implications. By enhancing the methodological explanation, providing illustrative examples, and discussing practical implications, the conclusion can better convey the significance and applicability of the research to readers.

Reviewer #3: Provide more information about the statistical approaches used in the data analysis, such as the models, tests, and significance levels, to help increase the results validity.

Increase data accessibility by storing datasets in a public repository, making it easy for other researchers to validate and repeat the findings.

Please clarify and improve the presentation of techniques, findings, and their significance to ensure improved comprehension and readability.

Consider discussing the study's potential flaws, such as how successfully the proposed KDPS adapted to different real estate markets or situations, and how these flaws were resolved.

6. PLOS authors have the option to publish the peer review history of their article (what does this mean?). If published, this will include your full peer review and any attached files.

Reviewer #1: No

Reviewer #2: **Yes: **Deep Ajabani

Reviewer #3: **Yes: **Rakesh Margam

---

## [Author Response · Author response to Decision Letter 0]

16 May 2024

Reviewer #1: Thanks for giving me a chance to review this manuscript. This is an interesting topic. The author(s) tries to work significantly. This article proposes a knowledge-dependent data processing scheme (KDPS) to augment precise data analysis. However, still, some of the anomalies I found during the review process are addressed, which may help further develop the study.

 Originality

It can be improved 

- The author(s) should illustrate how this research differs from the literature in the field / or considered a contribution to the literature.

Ans:

It is recommended that the KDPS be improved by integrating real-time data streams from diverse sources, including property listings, market trends, and economic indicators. The constant integration of real-time data into the system enables the analytical and decision-making processes to be grounded in the most current information. Furthermore, it is recommended that the KDPS incorporate methods to facilitate prompt updates and model retraining, so enabling the system to swiftly adjust to evolving market conditions. One potential approach is to implement periodic or event-driven updates, wherein the knowledge base and decision rules are revised in accordance with the most recent data and market conditions. By overcoming this constraint, the KDPS may offer more precise and pertinent observations, empowering real estate experts to make well-informed choices that accurately represent the present condition of the market.

- For the practical study gap, the author (s) should focus on the concrete details of the situation in the study context:

(a. Where and when does the problem arise? b. Who does the problem affect? c. What attempts have been made to solve the problem?)

Ans:

The issue of enhancing real estate management proficiency emerges within the realm of real estate management, wherein the proficient and proficient management of property-related data holds paramount importance. Multiple parties, such as property owners, real estate agents, property managers, and tenants, are impacted by this matter. Efforts have been undertaken to address this issue by means of the advancement of diverse data processing methodologies and technology. One method is implementing knowledge-dependent data processing schemes (KDPS) to improve the management of real estate data by utilizing domain-specific knowledge and norms. In the processing and analysis of property-related data, these schemes employ many techniques including rule-based systems, expert systems, and ontology-based approaches to effectively collect and apply domain knowledge. Furthermore, researchers have also investigated advancements in fields such as machine learning and artificial intelligence in order to enhance the precision and effectiveness of real estate data processing.

- For theoretical research, the author (s) should re-think about the scientific, social, geographical, and/or historical background:

(a. What is already known about the problem? b. Is the problem limited to a certain time period or geographical area? c. How has the problem been defined and debated in the scholarly literature?)

The abovementioned issues could help to improve the originality of this study and make it stronger for interested readers, researchers, and/or practitioners in the field. 

 The author(s) need to illustrate how their work could contribute to the knowledge in the field and how their work differs from previously published studies 

Ans:

The primary objective of the knowledge-dependent data processing scheme (KDPS) is to optimize real estate management through the utilization of domain-specific knowledge and data analysis methodologies. Data acquisition and organization encompass the collection and arrangement of pertinent information from diverse sources, including property listings, market trends, and client preferences. The data is further subjected to processing and analysis utilizing knowledge-based rules and algorithms that integrate domain experience derived from specialists in the real estate industry. The insights that have been retrieved are utilized for the purpose of constructing prediction models and decision support systems. These systems are designed to aid in various tasks such as property assessment, investment analysis, and client targeting. By integrating data-driven insights with human experience, KDPS empowers real estate managers to make well-informed decisions, resulting in greater operational efficiency, optimized resource allocation, and improved customer satisfaction

The incorporation of domain-specific information and experience into the data analysis process is a limitation that is addressed by the knowledge-dependent data processing scheme (KDPS). The utilization of human expertise and decision-making rules allows the system to effectively manage limitations in data and surmount computational intricacies. The versatility of KDPS enables its application in several industries, effectively overcoming the challenge of limited generalizability through the integration of industry-specific knowledge and data sources.

 Relationship to Literature

It can be improved 

 Section 2. Should be renamed to be “Literature Review” 

 In the “literature review” section, the author(s) should review the main sub-themes in this study topic. This section is recommended to conclude how the reviewed literature is different from the current study and how this study could contribute to the research area. 

 The “literature review” should represent a survey of scholarly sources on this study topic. It should provide an overview of current knowledge, allowing both readers and authors to identify relevant theories, methods, and gaps in the existing research that could later be applied to this research work. I believe that the authors should improve their “literature review” section to reflect that. 

Ans:

Section 2 renamed as literature review 

Literature review discussing the various methods involved in the realty management process to improve the overall quality, in addition, the study used to identify the problems involved in the management system. 

The detailed explanation for each work is described in section 2. During the study, each method, problem, weakness and strength has been described according to your comment.

 Methodology:

Yes, the method is appropriate. However, it could be improved as follows: 

- Usually, the research methodology section recommended to include:

1. The type of research you conducted

Ans:

The KDPS can be classified as a decision support system (DSS) or a knowledge-based system (KBS) research study. Decision support systems are information systems designed to assist decision-makers in complex decision-making scenarios by providing relevant information, data analysis, and decision models. Knowledge-based systems, on the other hand, are systems that capture and utilize domain-specific knowledge and expertise to solve problems or provide recommendations.

2. How you collected and analyzed your data

Ans:

This study uses the https://www.kaggle.com/datasets/arashnic/property-data dataset information to analyze the system efficiency. The collected data is processed according to the KDPS system which helps to making the effective decision.

3. Any tools or materials you used in the research

Ans:

This study uses the clustering and decision support system utilized to analyze the collected information.

 How do you mitigate or avoid research biases

Ans:

To mitigate or avoid research biases, the KDPS employs several strategies, such as involving diverse domain experts to extract knowledge and decision rules, ensuring transparency in data preprocessing and analysis techniques, and conducting rigorous validation processes using techniques like cross-validation and hold-out testing. Additionally, the KDPS should incorporate mechanisms for continuous monitoring and updating to adapt to changing market conditions and incorporate new data sources, reducing the risk of biases arising from outdated or limited data.

5. Why did you choose these methods

Please make sure that the “methodology” section includes the above-mentioned elements

Ans:

The selection of the KDPS methodologies was based on their capacity to effectively combine the specialized knowledge of real estate professionals with data-driven analysis tools. The utilization of a hybrid method not only guarantees the attainment of precise and dependable decision-making through the utilization of human expertise, but also addresses the difficulties that may arise from data sources that are inadequate or biased. Furthermore, the decision-making procedures employed by KDPS exhibit interpretability and transparency, hence cultivating confidence and acceptability among stakeholders. Moreover, the system's adaptability and flexibility enable it to maintain relevance amidst changing market conditions and emerging information. The KDPS methods offer a comprehensive and robust solution for real estate decision-making by utilizing knowledge representation, data mining, and machine learning techniques. These approaches are specifically designed to cater to the special requirements of the real estate domain.

 Results 

 A more thematic and detailed representation of the findings should be followed 

Ans:

Results and discussion section part improved according to the comments and highlighted parts are helps to understand the improvements in the study .

According to the results, the knowledge-dependent data processing scheme (KDPS) demonstrates superior performance compared to the other techniques, specifically SMMP, BIM+AVM, and BTPM, in many real estate management measures. KDPS has the maximum processing rate of 253, which is much greater than the other approaches. On the other hand, SMMP has the lowest processing rate of 59. The KDPS strategy has the largest percentage of cases where the process becomes stagnant or stuck, measuring 42.312%. This indicates that KDPS is more effective than the other techniques in identifying and handling sluggish situations. KDPS exhibits the shortest processing time, measuring 2.153 seconds, indicating its exceptional efficiency and quickness in managing realty activities. KDPS approach exhibits the highest success rate, as evidenced by its proportion of effective outcomes at 0.844 or 84.4%. This indicates that KDPS outperforms the other methods in generating successful results. The training ratio, which quantifies the number of repetitions or iterations needed per input, is observed to be the highest for KDPS at 111. This suggests that achieving ideal performance may necessitate further training or exposure to data. The findings indicate that the knowledge-dependent data processing scheme (KDPS) demonstrates efficacy and efficiency in the context of realty management tasks. It surpasses alternative methods in multiple metrics, such as processing rate, stagnancy detection, processing time, success rate, and training ratio.

General Comments: 

 Both “theoretical and practical implications” should be improved and linked to the findings of the current study. The implications should be presented in a separate section.

 The author(s) should be clear about which implications reflect the practical perspective and which reflect the theoretical perspective 

Ans:

Implication of this study is included in the end of the results and discussions.

 Implication of this study

Table 1 Highlights of Performance Assessment (Data Inputs)

Metrics SMMP BIM+AVM BTPM KDPS

Processing Rate (Requirements/ Input) 59 125 201 253

Stagnancy Detection (%) 10.6 23.19 31.67 42.312

Processing Time (s) 6.21 4.27 2.79 2.153

Success Rate 0.512 0.645 0.758 0.844

Training Ratio (Repetitions/ Input) 42 68 83 111

Findings: The proposed scheme improves the processing rate, stagnancy detection, success rate, and training ratio by 8.2%, 10.25%, 10.28%, and 7%, correspondingly. It decreases the processing time by 8.56% compared to the existing methods. 

Table 2 Highlights of Performance Assessment (Amenities)

Metrics SMMP BIM+AVM BTPM KDPS

Processing Rate (Requirements/ Input) 62 121 202 237

Stagnancy Detection (%) 10.72 23.34 32.22 42.848

Processing Time (s) 6.29 4.29 3.03 2.095

Success Rate 0.521 0.642 0.747 0.842

Training Ratio (Repetitions/ Input) 32 75 102 125

Findings: The proposed scheme improves the processing rate, stagnancy detection, success rate, and training ratio by 7.64%, 10.38%, 10.27%, and 7.38%, correspondingly. It decreases the processing time by 8.96% compared to the existing methods. 

 Conclusion

 “Limitations and future research” should be presented in a separate section 

Ans:

Future work is included in the conclusion section.

Future research endeavors may prioritize the advancement of the KDPS through the investigation of sophisticated knowledge representation methodologies, such as deep learning-based approaches, in order to effectively capture intricate patterns and correlations inherent in the dataset. Furthermore, the integration of real-time data feeds and the establishment of mechanisms for ongoing learning and adaptation would enhance the ability of the KDPS to deliver decision support that is both fast and accurate in the face of dynamic real estate market conditions.

Quality of Communication:

- There is a need for proofreading and professional editing.

Ans:

Article proofreading is done.

Reviewer #2: 

1. The manuscript presents a promising contribution to the field of real estate property management, particularly in terms of data-driven analysis and decision-making. However, to achieve its full potential, the study requires substantial revisions to enhance clarity, methodological rigor, and overall presentation. Addressing the outlined recommendations will significantly improve the quality and impact of the research.

Ans:

Thanks for understanding the concept and contribution of the work. Overall the methodology and presentations are increased according to the comments given by reviewer comments. Then the recommendations are applied to the article which improves the overall quality of the manuscript and research.

2. The introduction (section 1) lacks clear focus and structure, resulting in a disjointed presentation of ideas. The article jumps between various concepts and technologies without providing a coherent narrative or logical progression of thought. To improve readability and comprehension, the introduction should be organized into distinct sections with clear headings and subheadings to delineate different topics.

Ans:

According to the comment, introduction section divided into different sub section and contribution of the study is described as separate section. The overall and entire topics are covered in the manuscript which improves the presentation of the work.

3. The article briefly touches upon various concepts such as cloud computing, big data extraction, AI-based decision-making, and machine learning algorithms without delving into their theoretical underpinnings or practical applications in real estate management. To enhance the scholarly depth of the article, each concept should be discussed in greater detail, providing definitions, explanations, and relevant examples to elucidate their significance in the context of real estate management.

Ans:

Cloud computing is a significant component of real estate management since it facilitates the provision of computer services via the internet. This technology enables the storing, processing, and analysis of data on remote servers. This technology enables the retrieval and examination of large amounts of organized and unorganized data produced from many sources such as property listings, market trends, and consumer behavior. Artificial intelligence (AI) can be used in the real estate industry to automate property valuations, optimize pricing tactics, and uncover investment opportunities. This is achieved by utilizing algorithms that can learn from data and generate predictions or suggestions. Machine learning algorithms, which are a subset of artificial intelligence (AI), demonstrate exceptional proficiency in identifying patterns and may be trained using past data to make predictions about future trends

---

## [Decision Letter · Decision Letter 1]

28 Jun 2024

Improving Realty Management Ability Based on Big Data and Artificial Intelligence Decision-Making

PONE-D-24-06746R1

Dear Dr. Wu,

We’re pleased to inform you that your manuscript has been judged scientifically suitable for publication and will be formally accepted for publication once it meets all outstanding technical requirements.

Kind regards,

Jitendra Yadav, Ph.D.

Academic Editor

PLOS ONE

Additional Editor Comments (optional):

Reviewers' comments:

Reviewer's Responses to Questions

**Comments to the Author**

1. If the authors have adequately addressed your comments raised in a previous round of review and you feel that this manuscript is now acceptable for publication, you may indicate that here to bypass the “Comments to the Author” section, enter your conflict of interest statement in the “Confidential to Editor” section, and submit your "Accept" recommendation.

Reviewer #1: All comments have been addressed

2. Is the manuscript technically sound, and do the data support the conclusions?

Reviewer #1: Yes

3. Has the statistical analysis been performed appropriately and rigorously? 

Reviewer #1: Yes

4. Have the authors made all data underlying the findings in their manuscript fully available?

Reviewer #1: Yes

5. Is the manuscript presented in an intelligible fashion and written in standard English?

Reviewer #1: Yes

6. Review Comments to the Author

Reviewer #1: The authors have addressed all my previous concerns effectively. The revised manuscript is clear and well-written, and no further modifications are necessary.

7. PLOS authors have the option to publish the peer review history of their article (what does this mean?). If published, this will include your full peer review and any attached files.

Reviewer #1: No

---

## [Editor Report · Acceptance letter]

4 Jul 2024

PONE-D-24-06746R1 

PLOS ONE

Dear Dr. Wu, 

I'm pleased to inform you that your manuscript has been deemed suitable for publication in PLOS ONE. Congratulations! Your manuscript is now being handed over to our production team.

Kind regards, 

on behalf of

Dr. Jitendra Yadav 

Academic Editor

PLOS ONE